# Measurable Residual Disease Testing in Multiple Myeloma Following T-Cell Redirecting Therapies

**DOI:** 10.3390/cancers16193288

**Published:** 2024-09-27

**Authors:** Kevin Guanwen Shim, Rafael Fonseca

**Affiliations:** Division of Hematology and Medical Oncology, Mayo Clinic, Phoenix, AZ 85054, USA

**Keywords:** multiple myeloma, measurable residual disease, CAR-T, bispecific antibodies

## Abstract

**Simple Summary:**

Multiple myeloma (MM) is a blood cancer which classically has a prolonged course of remissions and relapses. Several new highly efficacious medications have been developed for MM in recent years, including some that utilize the body’s own immune cells to control the cancer. Alongside the development of these new medications has been the evolution of tests to track small amounts of (measurable) residual disease (MRD). Several MRD tests can now find and track as little as 1 residual cancer cell in 1 million in certain cases. Having no detectable MRD has been independently associated with improved cancer control and longer survival—here, we review how these tests might be used as a companion to new immune therapies for MM.

**Abstract:**

Several novel T-cell-based therapies have recently become available for multiple myeloma (MM). These T-cell redirecting therapies (TRTs) include chimeric antigen receptor T-cells (CAR-T) and bispecific antibodies (BiAbs). In both clinical trial and real-world data, these therapies have demonstrated high rates of deep clinical response, and some are now approved for second-line treatment for relapsed MM. The deep and sustained clinical responses these therapies are capable of inducing will require sophisticated response monitoring to provide meaningful information for patient care. Obtaining measurable residual disease (MRD) negativity has been validated as an independent positive prognostic marker for progression-free survival (PFS) and overall survival (OS) in both newly diagnosed and relapsed refractory patients with multiple myeloma. Assessment for MRD negativity was performed in all of the trials for FDA-approved TRT. Here, we summarize pertinent data for MRD assessment following TRT in MM and provide a rationale and structured framework for conducting MRD testing post TRT.

## 1. Introduction

In the treatment of MM, tracking a single malignant clone by the unique B-cell receptor rearrangements of plasma cells has been a powerful tool to better understand the disease. Use of MRD testing has evolved from an experimental descriptor of response depth to a key indicator of prognosis and treatment efficacy for patients with MM. Beyond its now well-defined prognostic value, attaining deeper and sustained remissions with MRD negativity holds promise for understanding the mechanisms that drive tumor resilience and persistence. The high degree of sensitivity inherent to MM MRD testing pairs well with the increasingly deep and sustained remissions induced by modern therapies. Specifically, recent advances in MM therapy have brought several therapies that reform autologous T-cells (CAR-T) or engage them via a linking antibody (BiAb) to rapid approval, with more anticipated in the future. In this article, we will refer to them as T-cell redirecting therapies. Attaining MRD negativity with these treatments has, and will, continue to be relevant for clinical decision-making, evaluation of new treatments, and further exploration of the biological mechanisms of disease eradication and escape. The published data on TRT in MM have already laid the foundation for how MRD testing will be utilized in the future to shape management decisions. Here, we aim to review the currently available data on MRD testing after T-cell redirecting therapies and outline a possible approach to utilizing MRD testing effectively in the future.

## 2. Current Clinical Use of MRD Testing in MM

### 2.1. Formal Guidelines for MRD Testing

There are numerous technical and logistical considerations defining the feasibility, availability, timing, and cost of different methods for MRD assessment [1,2,3,4,5,6]. Today, the most commonly used techniques to evaluate for MRD status are next-generation sequencing (NGS) and next-generation flow cytometry (NGF) assays. Some have also proposed the use of imaging MRD detection utilizing either MRI- or FDG-PET-based modalities in certain contexts, although imaging modalities lack the sensitivity of more modern direct disease measurements [7,8]. In the United States, the only Food and Drug Administration (FDA)-approved assay for MRD detection is the NGS assay clonoSEQ [9,10]. Internationally, the standardized NGF Euroflow panel is widely utilized [11]. 

Guidelines will be needed for MRD testing following TRT in MM but will have to rapidly evolve to reflect the growing body of knowledge on this subject. At the moment, there are no universally embraced set of timepoints or clinical contexts where MRD testing is recommended. Most pertinent to this discussion are the 2024 guidelines issued by the International Myeloma Working Group (IMWG) for CAR-T therapy [12]. Bone marrow biopsy is recommended at regular intervals beginning at 30 and 90 days post CAR-T treatment (whether testing at day 30 is perhaps too early in some patients remains an active area of study). MRD assessment is specifically recommended for patients achieving very good partial response or better (≥VGPR). Also relevant are the 2016 consensus criteria from the IMWG on MRD assessment. These recommendations establish a framework for utilizing MRD to evaluate the outcomes of modern MM treatments with an emphasis placed on the use of serial MRD evaluations. These guidelines introduced the new response criteria of sustained MRD negativity, which necessitates serial monitoring of MRD status. The key component of this response metric is a minimum duration of 1 year between negative MRD assessments. The second key element is the suggestion to evaluate MRD status at the time of achieving a complete or stringent complete response (≥CR) [2]. The response criteria also allow for further specification of the technique utilized by dividing response into flow, sequencing, or imaging plus MRD negativity. 

The National Comprehensive Cancer Network (NCCN) guidelines have adopted the IMWG MRD response criteria as above. The NCCN frames MRD testing as useful in certain situations for consideration during initial diagnostic workup and following autologous hematopoietic stem cell transplantation (auto-HSCT) [13]. Finally, in the 2022 statement from the American Society for Transplantation and Cellular Therapy (ASTCT), the utility and promise of MRD is acknowledged, as is the need for further data to establish clinically relevant timepoints for the use of MRD monitoring in the peri-transplant setting. The formal statement issued was that MRD should not currently be utilized to guide the timing of autologous transplant outside of a clinical trial [14]. Until a future when clinicians and societies have all the pertinent information in various clinical scenarios, physician discretion will be of paramount importance for timing MRD testing.

### 2.2. Prognostic Value of MRD Testing

The prognostic value of undetectable MRD in patients with MM has been validated in several meta-analyses and data were formally reviewed by the FDA’s Oncologic Drugs Advisory Committee (ODAC) in April, 2024 [15]. The outcome of this review was a recommendation to the FDA to support the use of MRD assessment as an endpoint in future clinical trials. Meta-analysis data were presented, demonstrating a strong statistical association between both progression-free (PFS)/overall survival (OS) and complete responders with MRD negativity at 9 and 12 months post-treatment [16,17]. This association was preserved in all populations studied (newly diagnosed transplant eligible and ineligible, and relapsed/refractory). Notably, none of the trials reviewed were evaluating MRD assessment in new TRT.

Two additional large studies have highlighted the impact of MRD testing across all phases of treatment for MM. In the first, patients with newly diagnosed MM (NDMM) had their MRD status evaluated following various treatment regimens. Across methodologies and treatment arms assessed, attaining undetectable MRD was associated with improvements in PFS and OS, whether or not a response by conventional criteria was attained. When patients with relapsed/refractory MM (RRMM) were studied in the second meta-analysis, similar findings were reported. This analysis additionally highlighted the ability of MRD negativity to serve as useful prognostic indicator, even in those with a higher cytogenetic risk [18,19].

Additional meta- and real-world analyses have supported these findings and also demonstrated the utility of MRD as a surrogate measure for PFS [20,21,22,23,24,25]. Indeed, MRD negativity is now one of the single most powerful prognostic features available to guide the management of MM patients (Table 1). These data are helping to refine the use of MRD testing in MM and bring the clinical use of MRD testing closer to the established frameworks used for other hematologic malignancies [26]. The implications of these findings in the context of the TRT now available has tremendous significance for patient care and is the central focus of this review.
cancers-16-03288-t001_Table 1Table 1Prognostication models for multiple myeloma.
GroupComparisonHR: PFSHR: OSMRD StudiesCavo 2022 [20]NDMM-TIE and RRMM≥CR + MRD-Neg vs. ≤VGPR or MRD Pos0.20-Munshi 2020 [19]NDMM-TEMRD Neg vs. Pos0.330.50NDMM-TIEMRD Neg vs. Pos0.320.40RRMMMRD Neg vs. Pos0.340.28Non-MRD StudiesR-ISS [27]NDMMStandard vs. HRCA0.540.49R-ISS (I vs. II)0.470.27R-ISS (I vs. III)0.280.1R2-ISS [28]NDMMR2-ISS (II vs. III)0.740.51R2-ISS (II vs. IV)0.490.24Hazard ratios (HRs) for progression-free survival (PFS) and overall survival (OS) for the studies and comparisons listed. The inverse of reported hazard ratios are adapted from those reported in the R-ISS and R2-ISS studies referenced above for ease of comparison. NDMM: newly diagnosed multiple myeloma. RRMM: relapsed refractory multiple myeloma. TE: transplant eligible. TIE: transplant ineligible.


### 2.3. Notable Examples of MRD Testing 

The MASTER trial is a prime example of response-adjusted clinical management based on MRD testing status and is a key foundational study for MRD response-adapted decision-making in clinical practice [29,30]. Similarly, the large, randomized phase III clinical trial, PERSEUS, has incorporated attainment of MRD negativity in patients with a complete response as a criterion to stop maintenance therapy [31]. Other notable MRD response-adapted trials currently underway include DRAMMATIC [32] and GEM2014MAIN [33], both of which are evaluating modern treatments in the maintenance therapy setting. MRD status is the primary endpoint in the GMMG-CONCEPT [34] and AURIGA trials [35]. Furthermore, several recent clinical trials now guiding treatment in clinical practice included MRD status as a key secondary endpoint and will inform the optimal strategy to evaluate for MRD [36,37,38,39,40,41]. Available ‘real-world’ data support the findings that have been reported in clinical trials, with MRD negativity serving as an independent prognostic indicator for both PFS and OS [42,43,44]

### 2.4. Technical and Logistical Considerations of MRD Testing

To date, the primary downsides identified for MRD testing are logistical. The high specificity of NGS in particular can also represent an underlying weakness, requiring identification of a suitable clone at baseline. Other factors such as disease heterogeneity, overall availability, and technical challenges associated with bone marrow biopsy collection (i.e., hemodilution and the invasiveness of the procedure) make standardization of the MRD testing process critical to its effective use. NGF is able to offset some of these challenges—namely, access and the inability to identify a usable clone—but comes with the challenges of lower sensitivity and the possible need for a greater number of cells to achieve the same sensitivity. All of the above factors must be taken into consideration in order to choose the most appropriate MRD testing methodology, which may vary based on the practice environment. The use of imaging MRD has less widespread use in clinical practice, but may remain useful in certain clinical situations, particularly in patients with plasmacytomas or bone disease. 

## 3. MRD Testing with CAR-T Therapy

Chimeric antigen receptor T-cells have been redirected, via viral transduction, to target malignancy-associated antigens. Engineering optimized constructs to create CAR-T therapies in MM is an active area of research [45,46,47,48,49]. CAR-T therapy has demonstrated excellent clinical outcomes for RRMM in published data to date. However, these are relatively new treatments for MM and answers to general questions about the durability of treatment response, temporal heterogeneity in response, and prediction of favorable response to treatment are especially pertinent to the discussion of optimal timing for MRD assessment and utility in the future. A summary of timepoints where MRD assessment occurred following treatment with TRT is presented in Figure 1. At the time of this writing, there are two B-cell maturation antigen-targeted (BCMA) CAR-T products available for patients with RRMM approved by the FDA and European Medicines Agency (EMA) (amongst other national regulatory agencies). Both products, ciltacabtagene autoleucel (cilta-cel) and idecabtagene vicleucel (ide-cel), have demonstrated similar clinical outcomes and rates of toxicity, and both have ongoing evaluations occurring. In April 2024, the FDA approved the use of both products for early use in RRMM after one line (cilta-cel) or two lines of therapy (ide-cel) [50,51]. Additionally, both series of trials (KarMMa and CARTITUDE) built upon promising early phase studies and collected MRD data, irrespective of what depth of clinical/traditional response to MM treatment was attained [52]. A summary of MRD testing results for FDA-approved TRT is presented in Table 2.
cancers-16-03288-t002_Table 2Table 2Summary of MRD data in CAR-T and BiAb trials.

ThresholdMRD Negative (%)MRD Evaluable
≥CR and MRD Negative (%)TechniqueTreated PatientsCAR-TIdecabtagene Vicleucel
KarMMa-1 [53]10^−5^33 (100)33
33 (100)NGS/NGF128
KarMMa-3 [54]10^−5^51-
-NGS225
10^−6^32-
-NGS
Ciltacabtagene Autoleucel/LCAR-B38M
CARTITUDE-1 [55]10^−5^56 (91)61
-NGS97
10^−6^39 (75)52
-NGS

CARTITUDE-2 [56]10^−5^7 (70)10
3 (43)NGS20
CARTITUDE-4 [57]10^−5^126 (88)144
-NGS176
CARTIFAN-1 [58]10^−5^39 (98)40
35 (90)NGF48
LEGEND-2 [59]10^−4^50 (68)74
50 (68)NGF74BiAbTalquetamab10^−5^11 (69)16
11 (100)NGF232Teclistamab10^−5^44 (81)54
30 (68)NGS165Elranatamab







MagnetisMM-1 [60]10^−5^13 (100)13
9 (69)NGS55
MagnetismM-3 [61]10^−5^26 (90)29
26 (100)NGS123Data available from published trials for each FDA approved CAR-T and BiAb for MM. The percentage of MRD-negative patients is reported as a fraction of those evaluable for MRD. The percentage of patients attaining a ≥CR and MRD negativity is reported as a fraction of those who were MRD-negative. Treated patients refers to the number of patients who actually received the TRT product in the respective trial. NGS: next-generation sequencing, NGF: next-generation flow cytometry.


In June of 2023, the Chinese National Medical Products Administration (NMPA) regulatory agency also approved the CAR-T product equecabtagene autoleucel (eque-cel), which is not currently approved for use outside of China. It is also targeted against BCMA but possesses a unique fully humanized receptor construct [62].
Figure 1A summary of timepoints where MRD testing occurred as reported in published trials for each FDA-approved CAR-T and BiAb for MM with accompanying references [53,54,57,60,63,64,65,66]. The left half of the figure highlights fixed timepoints after TRT where MRD was assessed in trials. The box in the upper right of the figure highlights response-modulated MRD assessment strategies utilized in pertinent trials. Test tube emojis represent a proposed set of standardized testing timepoints for all patients receiving TRT.
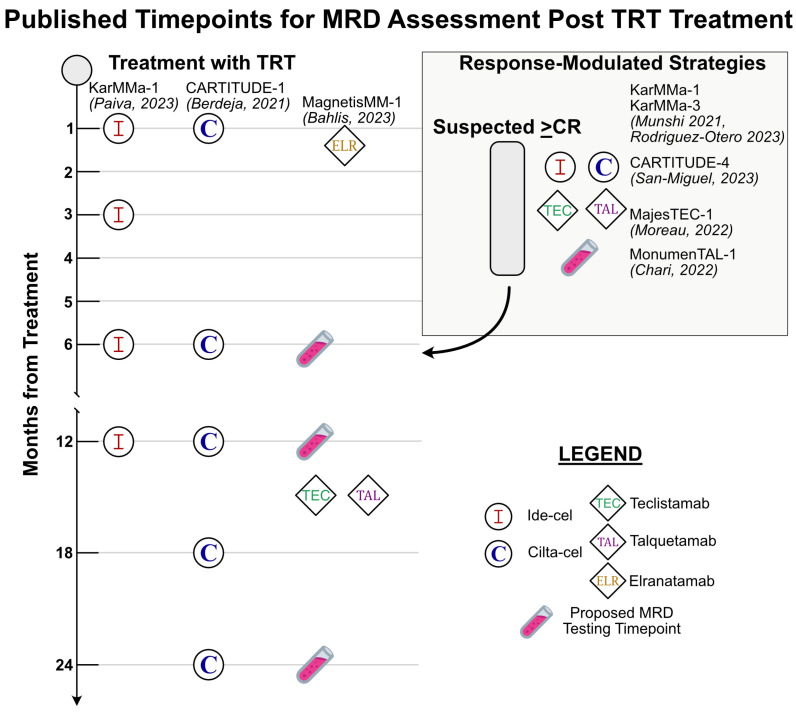



### 3.1. MRD Testing in Early Phase CAR-T Trials

For both FDA-approved products, MRD was assessed regularly during each phase of clinical trials. The CARTITUDE series of trials for cilta-cel assessed for MRD status at baseline, 28 days post-treatment, and then at regular intervals thereafter at 6, 12, 18, and 24 months post-treatment. In contrast, the early phase KarMMa-1 trial for ide-cel utilized a response-modulated approach to assess for MRD status. MRD negativity was reported for assessments performed within 3 months of achieving a ≥CR by the IMWG criteria. Long-term follow-up of the KarMMa-1 trial also included serial monitoring of MRD at 1, 3, 6, and 12 months irrespective of response. All studies in both series of trials utilized NGS with the clonoSEQ assay for evaluation of MRD with a sensitivity threshold of a minimum of 1 malignant cell per 1 × 10^5^ cells (i.e., threshold of 10^−5^). It should be noted that for regulatory purposes most trials report at this threshold, while the technology can routinely achieve a sensitivity of 10^−6^. The KarMMa-1 trial additionally utilized NGF for assessment of MRD and provided a direct comparison between techniques in an additional analysis. 

For the first-in-class ide-cel phase 2 trial (KarMMa-1), MRD negativity was reported within 3 months of obtaining ≥CR. All 33 patients evaluable for MRD were found to be MRD-negative [53]. In the phase 1b/2 trial for cilta-cel (CARTITUDE-1), all patients, irrespective of response status, were assessed for MRD by NGS at fixed timepoints. Here, 57 total patients were evaluable for MRD—including 22 patients that achieved a VGPR or less. Of the patients achieving a ≥CR, MRD was undetectable in 33 of 35 total evaluable patients [63]. In both trials, MRD negativity was ultimately assessed within approximately 3 months of CAR-T product infusion: the median time to response of a ≥CR was 2.8 months for ide-cel and 2.9 months for cilta-cel. The Chinese counterpart phase 2 trial for the LCAR-B38M product (which utilizes the same chimeric receptor construct as cilta-cel), CARTIFAN-1, had similar outcomes: 40 of 48 patients were evaluable for MRD and nearly all (39/40) achieved MRD negativity by NGS to a 10^−5^ threshold [58,67]. 

Building upon their initial findings, both KarMMa-1 and CARTITUDE-1 have reported further detailed outcomes at 12 and 27 months of follow-up, respectively. At a median follow-up duration of 27 months, the patients enrolled in CARTITUDE-1 with sustained MRD response for either 6 or 12 months had the best observed outcomes. Rates of PFS at 27 months of follow-up were 73% and 79% for patients with sustained response for 6 or 12 months, respectively. These responses compared favorably to the PFS rate of 55% noted in all patients at this same timepoint. Nearly all evaluable patients were able to achieve MRD negativity (91.8%) and a substantial percentage had sustained MRD negativity at 6 and 12 months (68% and 55% of evaluable patients, respectively). It should be noted that these figures represent overall small numbers of total patients (50 patients at 6 months and 44 patients at 12 months) and these observational findings were not designed for formal comparison [55]. 

For ide-cel, Paiva et al. conducted an elegant assessment of serial MRD negativity utilizing both NGS and NGF for patients enrolled in the KarMMa-1 trial. In contrast to the cilta-cel studies, rates of PFS were enumerated after landmark MRD assessments at 1, 3, 6, and 12 months. Beginning at 3 months following ide-cel infusion, obtaining a ≥CR with MRD negativity was associated with relative improvements in median PFS. Notably, MRD negativity with a response <CR did not demonstrate the same benefits. This may be driven by a subset of patients that experienced only transient MRD negativity in the first 6 months following TRT or the significant fraction of patients with EMD reported in this trial. Furthermore, the potential prognostic utility of sustained MRD negativity was demonstrated with improved rates of PFS in the patients never having evidence of MRD. Attaining MRD-negative status was associated with a hazard ratio for progression of 0.11 at 6 and 12 months in favor of those achieving MRD negativity [64,68].

In summary, in both the CARTITUDE-1 and KarMMa-1 trials, patients with sustained MRD for 6 or 12 months had better PFS outcomes than the patients who did not meet sustained MRD criteria. These data provide the strongest argument for the use of serial MRD evaluation following CAR-T therapy and support the idea of sustained MRD negativity having prognostic value in this specific context. Though data are descriptive at this point, it is interesting to note that patients in CARTITUDE-1 who attained sustained MRD negativity at 6 or 12 months had longer PFS than the patients achieving a sCR. Similarly, the KarMMa-1 data report that even if patients were able to attain a ≥CR, having detectable MRD at serial timepoints was (despite this degree of clinical response) associated with poorer PFS. Finally, long-term KarMMa-1 data provide a robust comparison between NGF and NGS methods for MM. A high rate of concordance was reported, with a variety of discrepancies noted because of the hemodilution of bone marrow samples utilized for NGS. Strikingly, they observed a fraction of patients that did not achieve a CR but still had undetectable MRD which had more favorable PFS and survival outcomes than the patients that had neither a CR nor MRD negativity. These results mirror those found in patients treated with non-T-cell-based therapies for MM and provide evidence to support the use of MRD testing as a response metric in patients treated with CAR-T. 

Eque-cel has also demonstrated a significant ability to induce MRD negativity. The phase Ib/II study FUMANBA-1 has reported sustained MRD outcomes at 6 and 12 months post-treatment. Patients had MRD assessed at a baseline and day 14 or 28, followed by 2-, 3-, 6-, 9-, 12-, 15-, 18-, 21-, and 24-month timepoints irrespective of disease status. MRD was assessed by NGF to a threshold of 10^−5^. At the most recent follow-up (12/2023), 88 patients had achieved MRD negativity, which was sustained in 58 of the 74 patients evaluable at 6 months (78.4%) and 32 of 43 patients (74.4%) evaluable at 12 months. These findings are correlated with prolongation in PFS for those achieving MRD negativity, though data are not yet finalized [69,70,71]. 

### 3.2. MRD Testing in Landmark CAR-T Trials

KarMMa-3 and CARTITUDE-4 are the landmark trials at the time of this writing for CAR-T therapy in MM. In each phase III trial, CAR-T therapy was compared to standard of care for RRMM. Both trials specify in their protocols a response-modulated approach to the use of MRD testing by NGS at the time of achieving a ≥CR. The CARTITUDE-4 trial additionally had serial assessments of MRD planned at 6-month intervals beginning at 1 month, and then yearly for a total duration of 2 years. 

For KarMMa-3, 51 of the 254 (20%) patients in the ide-cel arm were noted to have a ≥CR and MRD negativity; the number of total evaluable patients for MRD was not reported (the total number of patients with a ≥CR was 98). Only 1 patient of the 132 receiving a standard treatment regimen in the comparator arm was noted to be MRD-negative at any point (the total number of patients with a ≥CR was 7) [54]. This study reported MRD negativity as a negative MRD test within three months of attaining a ≥CR. In CARTITUDE-4, for the patients receiving cilta-cel, 87.5% of those evaluable for MRD status (126 of 144 evaluable) had undetectable MRD at any point during the study. In comparison, the patients receiving standard of care had an undetectable MRD rate of 32.7% (33 of 101 evaluable). Overall, 60.6% of cilta-cel-treated patients and 15.6% of patients in the standard of care arm achieved MRD negativity at any point during the study. These outcomes are, at the very least, correlated with the excellent clinical responses observed for these patients [57]. 

No studies have conducted a head-to-head comparison of currently available CAR-T products. Post-hoc indirect cross-trial comparisons have been published, providing a novel lens with which to view the predicted efficacy of CAR-T therapy, but neither study takes MRD status into account in their report [72,73]. Differences in MRD response rate between the two CAR-T products are hypothesized to be due to a higher fraction of patients with high-risk cytogenetics and a slightly higher median number of previous therapies in the ide-cel trials. Differences in the engineering of these products (cilta-cel is a bi-epitope-targeted construct) are also hypothesized to contribute to the observed differences. Given the relative novelty of CAR-T therapy, real world data are still limited. One of the earliest reports is of 159 patients treated with ide-cel at French centers included in the FENIX study, including 50 patients who did not meet the criteria for the KarMMa trials. MRD negativity was noted in 37 of 47 evaluated patients (79%) [74]. 

Poorer outcomes with extramedullary disease are a common thread amongst all trials and represent a place to begin seeking improvement. Rates of extramedullary disease (EMD) in the phase III trials were as follows: 61% (ide-cel) and 41% (cilta-cel). Serial imaging monitoring was performed where clinically appropriate in these and previous studies, but currently no imaging MRD data are available for review for any of the TRTs discussed above. Available data suggest overcoming extramedullary disease recurrence will be a challenge for TRT, as it has been for other treatment modalities. The retrospective data available have demonstrated EMD as an independent predictor of ORR and a trend towards worse rates of attaining MRD negativity in patients treated with ide-cel [75,76,77]. One hypothesis to explain this observation is that EMD may act as an antigenic sink or that ‘mass effect’ prevents penetration of immune effectors into tumors. In cases where EMD is the predominant manifestation of disease, MRD status assessed in the bone marrow may have a more limited role. Similarly, approximately 40% of patients in the treatment arms of both ide-cel and cilta-cel were noted to have high-risk cytogenetics. Further data will likely be forthcoming to more clearly demonstrate whether undetectable MRD following CAR-T treatment will be able to overcome the negative prognosis associated with high-risk cytogenetics as has been demonstrated with standard therapies.

### 3.3. Future of MRD Testing with CAR-T Therapies

Innovations in target selection, receptor engineering, and cytokine production for further CAR-T products continue to arise. Perhaps most notable is the MCARH109 product targeting G protein-coupled receptor, class C, group 5, member D (GPRC5D), which has demonstrated early safety and evidence of efficacy in patients with RRMM. In total, 8 of 12 patients treated at a safe dose level were noted to have undetectable MRD at some point during the study, despite all patients having penta-exposed disease (previously treated with 2 proteosome inhibitors, 2 immunomodulatory drugs, and an anti-CD38 antibody) and most having been treated with BCMA-directed treatments (including BCMA-targeted CAR-T) [78]. 

Another notable product is zevorcabtagene autoleucel (zevor-cel, CT053), which also targets BCMA. Phase IB data from this product show similar efficacy to other FDA-approved CAR-T therapies. Data presented from the ongoing LUMMICAR-2 trial assessing this product provide evidence of clinical efficacy and the finding that all patients obtaining a ≥CR were noted to be MRD-negative (11 patients, 78% of study population) [79,80]. Of note, the previously developed CAR-T product orvacabtagene autoleucel, as studied in the EVOLVE trial, is no longer under development but its BCMA-targeted construct is now being utilized as CC-98633/BMS-986354 as a novel less differentiated CAR-T product [81,82,83]. 

A full review of the developing CAR-T products is outside the scope of this review and has been discussed elsewhere [48,49,84]. We will highlight only that there are numerous promising strategies in development that will likely drive deep responses in patients with MM. For example, there are early data to suggest that previously developed CD19 CAR-T therapies in other hematologic malignancies may have a role in MM treatment. Current examples include trials demonstrating safety and efficacy in patients receiving CD19- and BCMA-targeted CAR-T therapies both simultaneously and sequentially [85,86]. Both of these trials are notable in that they recruited patients with either NDMM or those having received, on average, fewer lines of treatment as compared to earlier landmark CAR-T therapy trials. Both of these trials also observed most treated patients attaining undetectable MRD during the course of the study and that a deepening MRD response to treatment occurred over time. Preclinical work that will be of interest to follow includes CAR-T directed at CD38 and multiple simultaneous targets [87,88].

## 4. Bispecific Antibodies (BiAbs)

There are less data for BiAb therapies and the role of MRD at this time when compared to CAR-T therapies. There are currently two products targeting BCMA and one targeting GPCR5D approved by the FDA and EMA for RRMM. In the United States, all three products are approved for patients with RRMM that have received at least four prior lines of treatment and have been triple-class exposed (to an IMiD, PI, and CD38 targeting therapy). MRD status for these products has been assessed as an exploratory endpoint and its specific role following treatment with BiAb therapy remains to be defined. The number of available products is expanding rapidly and there are several other agents actively in development [89,90]. A summary of MRD testing for FDA-approved BiAbs is presented in Table 2 and the timing of MRD testing strategies is summarized in Figure 1.

### 4.1. MRD Testing in Early Phase BiAb Trials

The first two BiAbs approved in the United States were teclistamab, targeting BCMA, and talquetamab, targeting GPRC5D. The studies were designed similarly, with response-modulated assessments for MRD negativity in the early phase trials MajesTEC-1 (teclistamab) and MonumenTAL-1 (talquetamab). Both trials assessed for MRD at the time of suspected CR and 10 to 12 months post CR when possible. Both studies assessed MRD to a depth of 10^−5^ utilizing the NGS clonoSEQ assay (teclistamab) or institutional NGF (talquetamab). The median time to response was approximately 1 month in both trials and MRD assessment thus would have occurred in an approximately 1–4-month time window thereafter [65,66].

For teclistamab, 44 of 54 patients evaluable for MRD status in MajesTEC-1 (81.5%) were found to be MRD-negative. In total, 30 of these patients with MRD negativity attained a complete response or greater. Long-term data presented in 2024 at a median follow-up of 30 months suggest a durable response with a mPFS of 11.3 months and median duration of response of 21.6 months. In total, 48 of 58 evaluable patients (85%) achieved MRD negativity during treatment. Corresponding with these outcomes were promising metrics of sustained MRD negativity: 23 of 41 patients evaluable at 6 months (56%) and 14 of 36 patients evaluable at 12 months (38%) were reported [66,91].

For talquetamab, 16 patients had a ≥CR and were evaluable for MRD. In total, 11 of these patients (69%) were found to be MRD-negative. Long-term data for the results of the MonumenTAL-1 study were also presented in 2024 at an approximate median follow-up of 25 months. Varying dose levels are currently being evaluated and MRD negativity was not reported on, though high rates of ≥VGPR 55–59% were noted across all dose levels [65,92].

Elranatamab is a BCMA-targeted product that has been evaluated in the MagnetisMM series of trials. In both published trials, the focus was on MRD negativity in patients who attained a ≥CR. MRD status was assessed with NGS by the clonoSEQ assay. In the phase I MagnetisMM-1 trial, MRD status was assessed at approximately 1 and 3 months after treatment, with varying frequency thereafter. Of 21 patients with a confirmed CR or better, 13 were evaluable for MRD and all achieved negativity at some point during the study. In the phase II follow-up, MagnestisMM-3, 29 of 43 patients were evaluable for MRD and 26 of them (89.7%) were found to be MRD-negative at any point in the study [60,61]. 

### 4.2. Future of MRD Testing with BiAb Therapy

The ‘off-the-shelf’ nature and accessibility of BiAb therapy when compared to CAR-T continues to drive interest in whether sustained MRD can be attained with these treatments. The most robust data currently available for products under development are for the BCMA-targeting product linvoseltamab, with data published in 2024 from the phase I/II LINKER-MM1 trial. In total, 19 of 21 MRD-evaluable patients achieved MRD negativity from treatment to a threshold of 10^−5^ utilizing NGS. All patients evaluated for MRD were noted to have a ≥CR [93,94]. Similarly, cevostamab is a FcRH5-targeted BiAb that is currently recruiting for early phase trials. It has been granted orphan drug designation by the FDA and EMA for RRMM. No data are yet available on the ability of this agent to induce MRD negativity, but clinical results are promising, with nearly all patients responding to treatment [95,96]. 

Similar to those conducted for CAR-T therapy, there have been several innovative studies utilizing adjusted models to compare data from elranatamab and teclistamab clinical trials to patients receiving treatment in other non-BiAb-assessing clinical trials, but rates of MRD are not compared [97,98,99]. Additionally, real-world outcomes of BiAb therapy have been published, highlighting the overall generalizability of these therapies for patients not meeting stringent clinical trial criteria [100,101]. 

Of the many novel BiAbs currently under investigation, early data on the ability to attain MRD negativity are available for at least four products—all targeting BCMA: AMG 420, ABBV 383, AMG 701 (pavurutamab), and BMS-986349 (alnuctamab). MRD reporting for the first three products is limited to few patients (<10 in any given study) and utilizes a variety of MRD monitoring techniques at various thresholds [102,103,104]. Alnuctamab has reported 100% of patients enrolled in the study (14 patients) demonstrating undetectable MRD at approximately 1 or 4 months after treatment initiation [105]. A review of other promising BiAb and BiAb combinations actively being studied is beyond the scope of this article but is anticipated to continue demonstrating high effectiveness in attaining deep responses [106,107,108,109,110].

## 5. Timing of MRD Assessment

The various timepoints of MRD status assessment in pertinent TRT trials is summarized in Figure 1. In addition to these clinical data, many have already commented on when the optimal timepoint for MRD assessment may be. The position of the IMWG is that annual testing upon obtaining MRD negativity is a valuable response metric. Consensus in the field seems to be converging on the conclusion that serial monitoring of MRD status will hold more value than any single MRD testing timepoint. The obvious limitation of this argument is that taken to the extreme it becomes tautologic. Indeed, those with many points of sustained MRD negativity will fare better, but at which number of determinations monitoring can be safely discontinued remains of key importance. The IMWG’s T-cell therapy guidelines represent the first formalized recommendations for guiding MRD evaluation following these therapies. Other commonly discussed and utilized timepoints for MRD in current clinical practice include post-induction therapy, post-transplantation, and post-consolidation at clinically determined intervals during maintenance therapy [4]. The bulk of data currently available specific to TRT assess for MRD within the 3 months of attaining a CR, with serial monitoring thereafter. Data from the long-term follow-up of the KarMMa-1 and CARTITUDE-1 studies do provide evidence that sustained MRD negativity translates to a lower risk of progression, with the benefit becoming more pronounced with longer durations of MRD negativity at 6 and 12 months post-treatment. In this context, 12-month sustained MRD negativity has been proposed as a standard CAR-T response criteria [55,64,111,112].

### 5.1. MRD Testing Timing for CAR-T Therapies

There is a clear clinical rationale for checking MRD status after a ≥CR by the IMWG criteria is obtained, but there are also biological signals suggesting that MRD may have use at a stage of lesser response by traditional criteria. For example, several trials tracking serial MRD status have demonstrated that the kinetics of monoclonal protein decline may be delayed and traditional response criteria may not yet be met in a small percentage of patients that have no evidence of MRD due to the long half-life and elimination time of monoclonal antibodies [54,63,64]. The full clinical utility of attaining this degree of response (i.e., early MRD response with a conventional response to follow) is not yet fully understood, and data have only recently emerged for these response criteria following treatment with TRT. 

The pharmacokinetics of these T-cell products may provide a rationale for earlier testing for MRD response outside of conventional criteria. Both FDA-approved CAR-T products reported a median peak concentration between 11 and 13 days after infusion. Cilta-cel transcripts were reportedly not detectable in most patients at 6 months of follow-up, with detectable product found in patients for a median of 57 days (longest 631). Conversely, ide-cel was detectable in 59% of patients at 6 months and in 36% at 12 months. Though the overall number of studied patients is small, no direct association between the persistence of a detectable CAR-T transcript and a longer sustained response has been reported. Early single-cell sequencing data following treatment suggest that there may be a difference in survival detectable by changes in the bone marrow microenvironment at as early as 28 days after CAR-T therapy. However, the clinical applicability of these findings remains to be realized as the data also suggest that nearly all patients have a detectable tumor following CAR-T therapy at this early timepoint [113]. We have favored allowing extra time for infused CAR-T to exert an anti-MM effect and to measure MRD at day 100, but there are conflicting data, as exemplified below. 

There are some intriguing potential benefits to early testing for MRD after TRT. In a single-institution, real-world review of CAR-T outcomes at 1-month post-treatment, we explored the utility of MRD assessment at an early time stage in therapy. In total, 12 of 13 patients with detectable MRD at 1-month post-treatment continued to be MRD-positive at serial assessments. Furthermore, the rate of conversion from MRD positivity to negativity was low (5%). Early MRD assessment may thus be a specific testing modality for the prediction of treatment failure [114]. 

For cilta-cel, 93% of patients evaluable for MRD had achieved negative status at a median of 1 month following treatment; the median time to first response was the same for the majority (79%) of responding patients [63]. The median time to best response was approximately 2 months. Data from eque-cel are unique in the multiple serial NGF assessments conducted during the FUMANBA-1 trial at close intervals. This group published a remarkably short median time to MRD negativity of 15 days (range: 14–186) following treatment. The median peak concentration (12 days) and median duration of CAR-T detectability (approximately 12 months) are similar to those reported for ide-cel and cilta-cel [115]. 

These data suggest that MRD may represent a sensitive metric for assessing response to treatment at an early timepoint. Early control of disease may provide information about which patients will have a sustained response and which need to begin planning for subsequent therapy. This was highlighted in the KarMMa-1 study, where rates of CR were noted to be lower than MRD negativity at each serial fixed timepoint at which it was measured. Early MRD testing may mitigate the observed phenomenon whereby CR lags behind the attainment of undetectable MRD. In this light, MRD presents an attractive method for reliably evaluating response faster than conventional response criteria are able to. However, as reported in KarMMa-1, 25 of 53 (47%) patients also demonstrated MRD negativity 1 month after treatment with progression following thereafter [64]. Finally, early MRD testing may be beneficial in the planning for repeat CAR-T therapy or preparation for transplant or BiAb therapy, which have demonstrated efficacy following relapse after CAR-T treatment [116].

### 5.2. MRD Testing Timing for BiAb

For both talquetamab and teclistamab, response was noted to deepen over the course of treatment, with patients generally achieving their best response roughly 3 to 4 months following therapy. However, MRD-negative responses have been noted at early 1-month timepoints [60]. The pharmacokinetics for these BiAb therapies follow a potentially slower pattern of attaining a deep response than those of the CAR-Ts and appear to be more variable. Talquetamab and elranatamab reported reaching maximum concentrations in approximately 1 week. Teclistamab on the other hand was noted to reach a steady state only after several weeks and have a significant half-life of 15 days [117]. These preliminary results and knowledge of the pharmacokinetics of all approved BiAb therapies suggest that MRD response could reasonably be assessed 1 to 3 months following treatment initiation, as reported in most published trials. 

## 6. Future Directions for MRD Testing Following TRT

Even as a more precise role for MRD testing is defined in clinical practice, several novel technologies are being leveraged for the use of deeper and more refined MRD detection. Significant research has been conducted on other methods of MRD detection including allele-specific oligonucleotide polymerase chain reaction (ASO-PCR), mass spectrometry-based detection, monitoring of tumor-specific antigens, and circulating tumor DNA detection. There is great interest in the development of a peripheral blood ‘liquid biopsy’ assay for the detection of MRD for MM, as utilized for other hematologic malignancies [118,119,120]. The utilization of machine learning to predict MRD response from other clinically available parameters, mass spectrometry, and combination techniques are a few examples of novel approaches to MRD detection [121,122,123,124]. Extramedullary disease is a generally poor prognostic indicator for patients, which appears to have carried over into the T-cell redirecting therapy space. The use of imaging MRD assessments in particular may represent another avenue which will be needed for a complete evaluation of disease.

An exciting field of research in the context of TRT is the utilization of tumor-associated antigens as a method to track both MRD and evolving therapy resistance. As new therapies are developed, the biomarkers utilized to assess response are also expanding beyond the monoclonal protein excreted by malignant clones. Of the FDA approved therapies at the time of this writing, both CAR-T products and two BiAb products target BCMA. The kinetics of BCMA in response to treatment appear to be a promising measure of disease burden and may have potential to be used together with validated NGS and NFS techniques [125,126,127]. 

## 7. Discussion

The most pertinent question surrounding MRD assessment and TRT is how to improve awareness and usage by clinicians to guide clinical decision-making. Further data are expected from the above discussed trials on the long-term correlation of MRD status and clinical outcomes. The CARTINUE trial (NCT05201781), for example, will follow the initial cohort of patients receiving cilta-cel for 15 years. These results will help to shape the testing paradigm for MM patients receiving TRT and warrant ongoing thoughtful consideration. This is especially relevant given the approval of TRTs as second- or third-line therapies for RRMM. Beyond the goal of obtaining knowledge to refine our understanding of malignancy and the factors that drive it, MRD assessment is now clearly established as a useful measure to predict clinical outcomes.

### 7.1. Proposed Framework

Data from CAR-T trials thus far are promising for the future use of MRD testing as a prognostic indicator following TRT. This is especially true of the results of serial MRD testing. The results mirror the predictive value of MRD testing reported in the large-scale meta-analyses supporting its use in other treatment modalities for MM. Though such definitive information is not yet available for BiAb therapy in MM, the deep responses and promising clinical outcomes reported to date are suggestive of a central role for MRD testing following all TRTs. The IMWG criteria provide a practical and effective framework for approaching MRD testing following use of TRT. Referencing these guidelines and the data reviewed above, we believe that for all patients receiving TRT, MRD assessment should occur at evidence of a first response ≥CR, then at a frequency of every 3 to 6 months for the first year, and then yearly thereafter if there is no change in clinical status (Figure 1).

Utilizing MRD testing in this manner will achieve the following:Provide an informative early MRD status data point as early as 30–60 days following treatment.Align with testing conducted in most of the currently published TRT trials and take a step toward a widely standardized approach toward monitoring MRD.Provide objective response-mediated criteria for MRD testing following TRT that aligns with the strategy applied to MRD testing after other types of MM therapies.

We additionally support measuring the depth of response at 10^−6^ whenever possible. The rationale for this is most clearly illustrated in the example cohort of ten patients following CAR-T therapy with ide-cel who attained undetectable MRD at a threshold of 10^−5^ but were found to have detectable MRD at the 10^−6^ threshold. All of these patients were noted to have progression of disease shortly thereafter [64]. Furthermore, we advocate for the routine use of NGS methodologies due to the high degree of specificity, depth of detection afforded, and standardization.

A potential downside to the testing strategy outlined above is the possibility of false negative results for the patients that might not yet have achieved the traditional response criteria due to the delayed clearance of monoclonal protein, but actually have undetectable MRD. Further discussion and refinement of the use of traditional response criteria to trigger testing for MRD will need to occur in the future. There are examples of studies, including some evaluating outcomes from TRT, that have now reported on the situation where traditional response criteria are discordant with MRD testing [54,128,129]. In most of these reported cases, the situation is such that MRD negativity preceded a complete response. Testing at a later timepoint to deliver information to patients with more certainty is felt to be more desirable than relaying a potential false negative result. 

### 7.2. MRD Testing for Patient Care

We have previously advocated for day 100 MRD testing post-transplant as an important data point for clinical decision-making for patients [130]. The proposed framework to assess MRD status following TRT will be valuable for many of the same reasons. Though there may still be uncertainty about the precise impact of attaining MRD negativity for clinical outcomes after CAR-T and BiAb therapy, it will help to provide reassurance and guide patient-centered decision-making conversations. Is there a role for MRD testing in patients receiving TRT who are not noted to have a complete response or better? We feel that the evidence available strongly suggests that the answer to this question is yes. Certainly, the knowledge gained from these assessments will shape the future of MM treatment, but also a discussion about the information obtained from MRD testing (whether detectable or not) is felt to be beneficial for patients in the here and now. 

Thus far, perhaps the most significant benefit of utilizing MRD status to influence clinical care plans has been its use as a means to safely de-escalate patients from long-term maintenance therapy having a negative impact on quality of life. For CAR-T therapy, early treatment discontinuation is less of a concern than maintenance regimens in current clinical use. However, all BiAb products that are currently approved were studied with indefinite treatment until disease progression or treatment intolerance. MRD testing may become useful as a means to safely de-escalate these patients from ongoing therapy. Sparing patients the potential adverse impacts of treatment and detriments to quality of life from prolonged healthcare contact have been positive features for both types of TRT thus far. Particular interest has emerged in treatment with BiAbs for the elderly or those whose goals of care are not in line with potentially prolonged and intensive hospital admissions. The possible use of CAR-T in earlier lines of therapy and for more frail patients makes the ability to safely de-escalate treatment even more critical [131,132,133]. 

Aside from the primary consideration of patient care, there are several other factors strengthening the rationale for utilizing MRD testing following TRT. Given the timeframe associated with the production of CAR-T therapies, assessment of MRD at early timepoints may be particularly valuable for patients who may be re-treated with the same therapy. Both cilta-cel and ide-cel have demonstrated the utility of their products in re-treatment after failure and the BiAb therapies available have also been studied following other TRTs [56,116]. The median time from leukapheresis to infusion was reported as 49 (ide-cel) and 44 (cilta-cel) days in the phase III trials. The experience with the global COVID-19 pandemic has helped highlight the significance of accounting for these types of logistical considerations and the impact they may have on patients [134]. 

## 8. Conclusions

Cellular therapy drives a deeper response to treatment than many previously utilized therapies for MM, and thus an appropriate metric to measure and assess the outcomes meaningfully for patients with MM is critical. Though newer in the treatment model for MM, the data from patients who have received BiAb therapy are demonstrating similar depths of response. TRTs are felt to control MM by a completely distinct mechanism to any treatment before their kind. The mechanism of external cell-mediated killing, independent of antibody binding, cytokine, or internal cellular signaling, is felt to underlie the deeper and more robust responses that have been seen with TRT [135]. This mechanism of action represents a fundamentally distinct biological mechanism of action from the previous changes to the native immune system from the simple ratio of CD4:CD8 cells to the size of memory CD8 cell populations, speaking to a modification of the entire immunological signaling network in patients exposed to TRT [136].

Overall, these findings suggest a unique mechanism of achieving disease control and the need for a refined approach for monitoring response to disease. We contend that MRD status will provide an effective method to assess this response and provide room for growth, perhaps with multimodal strategies to guide the development of other cellular therapies. The ability to seek and destroy residual clones of tenacious disease is a remarkable advancement in the field of cancer therapeutics. We feel that MRD is the companion to these treatment advances that will help fully reap the benefits of these treatments for patients by allowing clinical teams to provide reassurance and meaningful forecasts beyond what is possible with classical metrics. It provides a means with which to drive care for MM toward a more proactive rather than reactive clinical treatment and decision-making paradigm. Understanding the biology of how these resistant clones are able to survive active immune targeting will additionally help guide future therapy options and advances.

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
