# Peer review of "Measurable Residual Disease Testing in Multiple Myeloma Following T-Cell Redirecting Therapies"

_cancers, 2024, doi:10.3390/cancers16193288_

Round 1
Reviewer 1 Report
Comments and Suggestions for Authors
The authors report measurable residual disease testing in multiple myeloma following T cell redirecting therapies.
1. The meaning of Figure 1 is not clear. At least the Figure legend should be described in detail including abbreviations.
2. In Figure, the authors should present representative NGS and NGF data for MRD-negative cases.
Reviewer 2 Report
Comments and Suggestions for Authors
This is a well executed narrative review on the current role of MRD in patients with multiple myeloma receiving T-cell redirecting therapies including CAR T-cells and bispecific antibodies. The article is well written and the authors discuss the limitations and future directions. The only drawback is that IMWG has recently published consensus papers in the same field and, therefore, the novelty of this review is very limited, taking into consideration that the primary data available on this issue are rather limited.
Reviewer 3 Report
Comments and Suggestions for Authors
Dear Authors,
The paper of Shim KG et al. is concerned with measurable residual diseases (MRD) testing in multiple myeloma (MM) following T-cell redirecting therapies (TRT), id est chimeric antigen receptor T-cells (CAR-T) and bispecific antibodies (BiAb). The Authors clearly presented current methods of MRD measurement referencing papers publishing within the last 5 years. References are appropriate and only a few of referencing papers are older than 5 years. Summery of the newest trials is the field of MM MRD following TRT is strong adventage of the paper, so it should be publish as fast as possible. The manuscript is relevant for the field and presented in well-structured manner what aids a reader in finding appropriate informations. The gaps in knowledge are well identified and will set goals in future trials. The conclusions are consistent with the evidence and presented arguments.
I have only two comments:
1. There is no need to repeat many times abbreviations of the same word or phrase.
2. I would like to point out strongly that it is not chargé against the Authors because they objectively presented current knowledge in the field. In the lines 214-223 the paper of Paiva et al. is citated and the reader’s attention is paid to sentence: „Notably, MRD negativity with a response <CR did not demonstrate the same benefits” (line 219). I suggest the Authors to comment why there are not the same benefits and what it means – maybe it is false negative MRD because of method limitation. Afore in text above (lines 136-146) the Authors describe NGS and NGF, but it seems to be not enough clear for a reader which method is better and why and what clinical implications are or may be. In my opinion it should be describe more widely, all the more the Authors conclude „We advocate for the routine use of NGS methodologies for the reasons addressed earlier in the article (line 551) – not clear. Because MRD testing has the predictive value in MM treatment, including CAR-T and there is not enough information available for BiAb so it is strong need to state definitions. It is worth noting that the Authors in the basis of their own research, clinical trials and meta-analyses suggest MRD testing time points during MM TRT (lines 537-539).
Kind regards,
Reviewer
Reviewer 4 Report
Comments and Suggestions for Authors
The manuscript contains a comprehensive review of the current clinical use and future directions for minimal residual disease testing in multiple myeloma following T cell redirecting therapies, inclusing CAR-T and bispecific antibodies. The authors highlight the methods used, the prognostic significance of MRD, summarize clinical trial data, and propose a structured framework for MRD testing post T cell redirecting therapies. This article is of interest for readers of Cancers and can be accepted after a minor revision.
The article is well writen.
Minor suggestions:
- Please revise Figure 1, make it more clear, current version is not understandable
- Double check grammar and spelling.
Comments on the Quality of English LanguageEnglish quality is good
Round 2
Reviewer 1 Report
Comments and Suggestions for Authors
none